# Biomimetic Implant Surfaces and Their Role in Biological Integration—A Concise Review

**DOI:** 10.3390/biomimetics7020074

**Published:** 2022-06-06

**Authors:** Mariana Brito Cruz, Neusa Silva, Joana Faria Marques, António Mata, Felipe Samuel Silva, João Caramês

**Affiliations:** 1Universidade de Lisboa, Faculdade de Medicina Dentária, Unidade de Investigação em Ciências Orais e Biomédicas (UICOB), Rua Professora Teresa Ambrósio, 1600-277 Lisboa, Portugal; jmarques2@campus.ul.pt (J.F.M.); admata2@campus.ul.pt (A.M.); 2Universidade de Lisboa, Faculdade de Medicina Dentária, Unidade de Investigação em Ciências Orais e Biomédicas (UICOB), LIBPhys-FTC UID/FIS/04559/2013, Rua Professora Teresa Ambrósio, 1600-277 Lisboa, Portugal; neusa.silva@edu.ulisboa.pt; 3Cochrane Portugal, Instituto de Saúde Baseada na Evidência (ISBE), Faculdade de Medicina Dentária, Universidade de Lisboa, Avenida Professor Egas Moniz, 1649-028 Lisboa, Portugal; 4Center for Microelectromechanical Systems (CMEMS), Department of Mechanical Engineering, University of Minho, 4800-058 Guimarães, Portugal; fsamuel@dem.uminho.pt; 5Bone Physiology Research Group, Faculdade de Medicina Dentária, Universidade de Lisboa, Rua Professora Teresa Ambrósio, 1600-277 Lisboa, Portugal; carames@campus.ul.pt

**Keywords:** implant surface, osseointegration, dentistry, oral surgery, oral rehabilitation, biomimetic, dental implants

## Abstract

Background: The increased use of dental implants in oral rehabilitation has been followed by the development of new biomaterials as well as improvements in the performance of biomaterials already in use. This triggers the need for appropriate analytical approaches to assess the biological and, ultimately, clinical benefits of these approaches. Aims: To address the role of physical, chemical, mechanical, and biological characteristics in order to determine the critical parameters to improve biological responses and the long-term effectiveness of dental implant surfaces. Data sources and methods: Web of Science, MEDLINE and Lilacs databases were searched for the last 30 years in English, Spanish and Portuguese idioms. Results: Chemical composition, wettability, roughness, and topography of dental implant surfaces have all been linked to biological regulation in cell interactions, osseointegration, bone tissue and peri-implant mucosa preservation. Conclusion: Techniques involving subtractive and additive methods, especially those involving laser treatment or embedding of bioactive nanoparticles, have demonstrated promising results. However, the literature is heterogeneous regarding study design and methodology, which limits comparisons between studies and the definition of the critical determinants of optimal cell response.

## 1. Introduction

In recent years, dental implants have been widely used to replace missing teeth in order to restore the natural functions of the stomatognathic system through appropriate mechanical properties, stability, adequate bone integration, and regeneration [1,2,3].

These medical interventions have progressively changed to meet patient needs and improve their quality of life. However, research has focused on finding simpler, faster surgical procedures that allow for multiple restorative options with improved aesthetics and precise prosthetic components that offer procedural safety [4,5,6].

The first documented dental implant dates back to 600 AD. Since then, various materials have been tested as dental implants, including gold, silver, porcelain, and other materials [7]. Professor Per-Ingvar Brånemark’s study in 1950 demonstrated that titanium structures could be permanently incorporated into the bone such that this interface could only be separated by fracture [8]. In this context, the term osseointegration was introduced to describe a direct structural and functional connection between living bone and the surface of an implant without the intervention of soft tissue [8,9].

Bone is a complex connective tissue composed of a mineralized matrix (30%), which provides mechanical strength, and an organic matrix (70%), providing elasticity and flexibility [10]. It is a dynamic structure that undergoes a modelling and remodelling process and plays an important role in microfracture healing and skeletal adaptation. Bone tissue consists of four cell types, two of which, osteoblasts and osteoclasts, are involved in bone resorption and bone apposition dynamics triggered by molecular stimuli, which in turn leads to the differentiation of mesenchymal stem cells (MSCs) [10,11].

The extracellular matrix (ECM) forms a structural architecture that surrounds cells and consists of four main components, fibers, multi-adhesive proteins, proteoglycan and non-proteoglycan polysaccharides, which influence the behavior of the resident cells. The cellular interactions with the extracellular matrix are a complex network that is critical to maintaining tissue homeostasis. Besides maintaining self-renewal, the ECM also regulates the differentiation ability of undifferentiated cells by modelling the activity of growth factors [11,12].

Research has focused on the long-term survival rates of dental implants [13,14], with osseointegration being the main criteria for clinical success [15]. Originally described by Professor Brånemark, the phenomenon of osseointegration is a consequence of a series of cellular and molecular events that begins immediately after implant placement due to the interactions between the implant, biological fluids and peri-implant tissues [16,17,18,19]. The osteointegration process can be divided into three phases:(1)The initial tissue response;(2)Peri-implant osteogenesis;(3)Peri-implant bone remodelling.

(1) Initial tissue response: The initial process begins with the preparation of the drill hole and implant placement. From 0 to 4 h after surgical trauma, calcium ions and plasma proteins adhere to the implant surface [11]. The surgical trauma leads to an inflammatory process through activation of the complement system and neutrophil infiltration, resulting in growth factor and cytokine release. Depending on the transmitted stimuli, macrophages differentiate into two different phenotypes: M1—pro-inflammatory or M2—anti-inflammatory. Clot formation, platelets, and the fibrin matrix serve as the scaffold for the migration, proliferation, and differentiation of leukocytes and mesenchymal cells. This process takes place 1 to 3 days after implant placement [20,21,22].

(2) Peri-implant osteogenesis: After 3 to 4 days of surgery, the angiogenesis activity and the reorganization of a blood clot in the granulation tissue of the cancellous bone are more evident, while a blood clot with a high number of erythrocytes and fibroblast-like cells is still present in the area adjacent to the implant. From 7 to 14 days, the MSCs differentiate into osteoblasts and produce a fibrillar noncollagenous extracellular matrix rich in calcium, phosphorus, osteopontin and bone sialoprotein, designated primary bone tissue [11,23].

(3) Peri-implant bone remodelling: The final stage of healing takes place at 2 weeks. It is characterized by an increase in primary bone apposition in direct contact with the pristine bone and implant surface [11]. The first signs of bone remodelling also appear in the primary osteoid. Osteoclasts direct the remodelling process from immature bone to highly mineralized lamellar bone by binding to the mineralized collagen matrix and creating a zone where the bone is deposited directly on the implant surface. While after 3 months of implant placement, there may be lamellar and non-lamellar bone around the implant, the process of osseointegration may take 1 year or more to complete [17,24,25].

Currently, an implant is considered osseointegrated when there is no relative movement between the implant and the bone and no symptoms under a loading force [16,17,26]. This biological fixation process is a prerequisite for any implant-supported prosthesis and its long-term success [16,19]. Although osseointegration has been extensively demonstrated from a histological and clinical perspective, our understanding of this biological process is still limited. However, some parameters that can influence the osseointegration process have already been described in the literature. These factors are related to the implant (design, shape, length, diameter, and others), the host (implant sit, bone cellularity and density, systemic diseases, and others), the surgery (tissue damage, soft tissue healing ability, speed perforation and others), healing time and exerted loads [17,27].

The success of osseointegration of dental implants has been reported in many studies; however, a systematic review of the Cochrane database found insufficient evidence for the superiority of any particular type of implant feature or system [2]. Despite all the factors that can affect osseointegration, in this work, we will focus mainly on factors related to the materials and biomimetic properties of implant surfaces (Figure 1).

## 2. Search Strategy and Data Retrieval

The PubMed, LILACS, Web of Science, and Cochrane Library databases were searched to identify relevant studies on the role of surface modifications in biological responses of hard and soft peri-implant tissues to dental implants. The following search terms were used and combined for the initial literature search: dental implant, implant surface, roughness, coating, bioactivity, bioactive, functionalization, zirconia, titanium, poly-ether-ether-ketone, osteoblast, fibroblast, and biological response. Pre-clinical (in vitro and animal studies) and clinical studies relevant to the topic were selected, and studies without surface characterization methods were excluded. Limits were articles published in the last 30 years in English, Spanish and Portuguese idioms. Study selection was made independently by two authors, and any disagreements were resolved by discussion. In addition, we manually checked the reference lists of the articles to identify other potentially relevant publications.

## 3. Dental Implants Base Materials

From a chemical point of view, dental materials can be divided into three large groups: metals, ceramics, and polymers. Furthermore, based on their biological response, these biomaterials can be classified into three main types: biotolerant, bioinert and bioactive [5,27]. The different biocompatibility underlines that none of the materials used in the dental practice is completely biologically acceptable. In this regard, artificial structures must be selected to minimize the adverse biological response while ensuring adequate function.

Scientific terms such as osteoinduction and osteoconduction involved in processes related to osseointegration are commonly found in the literature [10]. Despite the success of osseointegration reported in many studies, peri-implant diseases have also been reported over the years as a pathologic inflammation affecting a significant number of dental implants. These events underline the importance of new materials and improved implant surfaces. An ideal implant surface should exhibit excellent osteoconductivity and biocompatibility, improving peri-implant wound healing and osteogenesis and resulting in faster and more predictable osseointegration [27,28]. In addition to accelerating the bone healing and osseointegration process, which can enable an early loading of the dental implant, it would provide more predictable outcomes in compromised scenarios. The ideal material should also have high mechanical strength, excellent aesthetics, and a low affinity for bacterial adhesion.

### 3.1. Titanium

Pure titanium (Ti) is a transition metal with atomic number 22, a melting point of 1668 °C and a boiling point of 3287 °C. Upon contact with atmospheric air, Ti forms a thin layer of titanium oxide covering its surface; this mechanism is unique to titanium, silicon and zirconia. [29] The layer is more extensive when titanium implants are exposed to biological tissue, giving them excellent biocompatibility [17,29]. Titanium can be used as a pure metal or as an alloy containing other metals such as vanadium, aluminium, niobium, iron, magnesium or zirconium. According to the American Society for Testing and Materials (ASTM), there are 4 Ti classes, depending on the amount of oxygen, nitrogen, hydrogen and carbon introduced during the purification process. There is also Grade V, which is a titanium aluminum alloy with 4% vanadium (Ti6AL4V) [30].

Commercially, Ti is most commonly used for dental implants, particularly in the IV grade, which is the highest strength of the various grades of pure titanium, excluding alloys [31]. Titanium and its alloys are the material of choice for the manufacture of dental implants due to their excellent biocompatibility, corrosion resistance, elasticity modulus and excellent mechanical properties [8,17,32,33], which are important for long-term implant success [19]. Several studies have therefore reported high success rates of titanium implants. Disadvantages such as hypersensitivity reactions, the difference in the elasticity modulus, wear resistance, electrical conductivity and the grey colour of this material are all of concern. Alternative materials have been developed to provide biological stability with better or comparable mechanical properties [32] to improve the biological and mechanical properties of these dental implants.

### 3.2. Zirconia

Zirconia (ZrO_2_) is a crystalline oxide form of zirconium, a transition metal with atomic number 44. Structurally, zirconia is a polymorphic material that occurs in three forms: monoclinic, tetragonal, and cubic. During the cooling process, some microcracks may appear. To avoid this, oxides such as magnesium oxide (MgO), yttrium oxide (Y_2_O_3_), calcium oxide (CaO) and cerium oxide (Ce_2_O_3_) are added to keep the tetragonal structure at room temperature and control the stress [34,35].

Tetragonal polycrystalline zirconia (TZP) is a zirconia-based ceramic predominantly in the tetragonal phase and generally stabilized with yttrium oxide (3–6% by weight), referred to yttria-stabilized polycrystalline tetragonal zirconia (YTZP). The opaque and white colour of YTZP, together with high fracture strength, flexural strength, thermal stability, low thermal conductivity, chemical resistance, biocompatibility and low affinity for bacterial colonization, renders this material a good candidate for implant dentistry. Overall, different studies have found that zirconia and titanium have similar bone tissue integration [35,36]. However, despite technological advances in the manufacture of ceramics, the mechanical behavior of zirconia has limitations, namely the ageing process. This is a process involving the degradation of zirconia at low temperatures that leads to the formation of cracks in the ceramic and subsequently fracture [35].

A recent systematic and critical review suggested that zirconia implants are a promising alternative to titanium based on its superior soft tissue behavior, biocompatibility and aesthetics while maintaining the same osseointegration ability as titanium [36,37]. The literature indicates that treated zirconia surfaces exhibit better or similar clinical bone-to-implant interface outcomes compared to equivalent titanium surfaces [35,37,38]. However, other systematic reviews show contradictory results favouring the titanium surface’s behavior [2,39]. In these reviews, the evaluation of surface parameters was not considered as a variable influencing material behavior. Therefore, these findings need to be confirmed by integrating adequate characterization of the surfaces.

### 3.3. Polyether Ether Eketone

Polyether ether eketone is an organic polymer with a wide range of applications in the medical industry due to its excellent biocompatibility, radiolucency, chemical resistance, low density, and chemical properties similar to human bone [40]. PEEK competes with many metals and alloys and has been recognized as an alternative to the systematic use of metal alloys for a significant number of biomedical applications, including dental, orthopaedic and cardiovascular devices [41,42]. Its versatility, biocompatibility, chemical resistance to biodegradation, and aesthetic properties make this material an interesting polymer for dental implants [24,43]. In addition, this material has physical and mechanical properties similar to human bone, making its use in orthopaedics, traumatology, and spinal implants to minimize stress and consequently reduce bone resorption common [44,45,46].

PEEK has been shown in the literature to improve bone and soft tissue behavior such as early cell adhesion, viability and proliferation, all of which are linked to increased surface wettability. Furthermore, compared to a smooth surface, porous PEEK surfaces, such as titanium, promote osteoblast proliferation and differentiation [47,48]. These findings show that chemically inert materials such as PEEK do not promote a fibrous response. However, it can be modified to improve biocompatibility, bioactivity and cellular behavior, with topography playing a central role in these mechanisms [48].

## 4. Biomimetic Surface Properties

Several factors can affect the long-term success of dental implants, such as implant surfaces that play a key role in their longevity [19]. As previously described, changes in implant surface parameters have been reported in the current literature, e.g., biomaterial interface with peri-implant tissue and cellular behavior that can significantly affect the speed and resistance of osseointegration [17]. Physical and chemical properties, such as chemical composition, topography, roughness, wettability or contact angle, are the main surface properties [4,5,19].

### 4.1. Topography

The implant geometry has continuously changed and evolved over the years. Numerous reports have shown that the macro geometry of implants can affect the osseointegration process, such as good primary stability, implant sealing, and maintenance of marginal bone level [49,50,51].

The surface topography can be divided into three levels according to the scale: macro, micro and nano. Macrotopography is defined in a scale range from 10 μm to mm, and is found in most implants commercially available today with a cylindrical shape and thread design, which may play a key role in increasing implant stability [19,27]. In terms of microtopography, the scale range is 1–10 μm, which appears to accelerate and increase bone-to-implant contact, maximize adhesion between the mineralized bone and the implant surface, and provide more predictable long-term clinical outcomes [52,53]. While the scale range defined for nanotopography is between 1 and 100 nm, it is believed to play an important role in protein adsorption and cell adhesion. Most of these studies are performed in preclinical models that lack clinical validation since their exact function in vivo is unknown [54,55].

It is now known that surface topography is one of the key biomimetic factors that can directly affect the proliferation, structure, and alignment of human cells and their function and is also considered to be a critical determinant of cell adhesion [56,57,58]. However, there is no consensus on which physical topography or characteristic dimensions might be relevant for biomedical applications [49,59,60].

### 4.2. Roughness

Rough implants affect the response of osteogenic and inflammatory cells by increasing bone-to-implant contact and overall clinical success, with faster healing rates and potential for earlier loading times [4,11,61]. Several researchers have expressed interest in the directionally rough implant surface, particularly in animal studies that have shown superior osseointegration of rough surfaces compared to smooth or machined surfaces [55]. However, depending on the method used, roughened surfaces with different topographical properties can be generated, which can be an issue in terms of the definition of rough or smooth surfaces.

The three most commonly used methods to measure implant surface roughness are: contact profilometry, optical profilometry, and contact atomic microscopy [62,63,64]. Various parameters such as Ra and Rz can be used to assess surface roughness. Ra is an arithmetic mean between the highest and lowest points on the surface, and Rz is calculated by measuring the vertical distance between the highest and lowest points on the surface [55].

In 2009, Wennerberg and co-workers presented different ways of classifying surface roughness [55]: smooth or machined (Sa < 0.5 μm), minimally rough (Sa 0.5–1 μm), moderately rough (Sa 1–2 μm) and rough surfaces (Sa ≥ 2 μm). These findings led to increased research related to hard and soft tissue behavior in titanium and zirconia dental implants [65,66,67,68]. Most of them found that better bone cell behavior was associated with an increase in surface roughness [67,69,70]. Some studies on soft tissue response show somewhat contradictory results, with greater proliferation and spread of fibroblast cells on smooth surfaces [65,68,71]. A literature review of in vitro studies on this subject concluded that regardless of the implant surface distinctions that should be made, titanium or zirconia surfaces with micro rough topography resulted in increased osteoblastic differentiation, collagen type I production, bone matrix protein expression, and cell-matrix layer mineralization [72]. Other studies claim that surfaces with micrometer scale roughness appear to create a favourable osteogenic environment and can influence the differentiation of MSCs’ cells towards an osteoblastic phenotype [73,74]. In addition, literature on surface modification in osteoblast proliferation suggests that porous PEEK surfaces show an increase in osteoblast proliferation and differentiation compared to smooth surfaces, as observed on titanium surfaces [47,48].

Currently, numerous studies on dental implant roughness have found that, regardless of the base material, a moderately rough surface is ideal for optimizing osteogenic response [11,17,75,76]. Some benefits of dental implant surface roughness include primary stability and long-term mechanical stability. However, the greater risk associated with increasing surface roughness is the predisposition to bacterial colonization and consequently an increased risk of peri-implantitis. Some preclinical animal studies suggest that moderately rough surfaces may be more prone to the progression of peri-implant disease than others [77,78]. Therefore, a systematic review based on human clinical trials found no evidence of increased susceptibility to peri-implantitis for moderately rough surfaces [79]. A 5-year prospective, multi-center, randomized, controlled clinical trial comparing a hybrid implant to a moderate roughened implant and the results also showed no differences in the incidence of peri-implantitis [80].

### 4.3. Wettability

Surface wettability is believed to be a prerequisite property affecting biomaterial biological response. The most commonly used techniques to measure surface wettability are contact angle measurements, which measure the interaction between biological fluids, cells and the biomaterial surface. In general, contact angle measurements of titanium surfaces range from 0° (hydrophilic) to 140° (hydrophobic) [24,81]. The biomaterial surface wettability is regulated by surface properties such as chemical composition and topography. It can affect four main aspects of the biological system: 1—protein and other macromolecules adhesion to surfaces; 2—hard and soft tissue interactions with surfaces; 3—bacterial adhesion and biofilm formation; 4—clinical osseointegration rate. Therefore, the hydrophilic properties promote protein and cell adhesion and interactions of body fluids and may increase tissue healing kinetics [82,83,84]. Moreover, microbiome control may influence bacterial adhesion to implant surfaces through approaches involving probiotics, postbiotics and natural compounds, which should be considered in further studies [85,86,87].

Besides that, while a large number of studies emphasize the role of surface topography in biological response, very few studies have evaluated the wettability, surface energy and chemical composition of dental implants [88,89,90]. Since different techniques to characterize surface parameters were used among studies, it is difficult to establish direct comparisons. Today we know that surface modifications on a micro- and nanoscale, coupled with surface wettability, can modulate the biological response at the cellular adhesion level [83,91]. However, as previously noted, to date, there is no evidence supporting a superior implant surface for all steps of osteogenesis [2,92].

## 5. Implant Surface Modifications

Implant surfaces have been modified in various ways to improve osseointegration and the biological process. In clinical practice, there is a strong need for an implant that increases osseointegration, with reduced waiting time from insertion to implant loading, especially in areas with low bone density and low primary stability or in patients with systemic diseases that can affect the bone healing process [11,93,94].

Recently, dental implant manufacturers have been focused on surface modifications to mimic the characteristics of bone ECMs, which are considered to be the most effective way to improve the speed and quality of osseointegration [95,96,97]. To meet these needs, companies have developed a large number of implant surfaces with complex topographies and varying degrees of roughness and chemical compositions. These modifications can create macro, micro, and nanostructures of various shapes on the biomaterial surfaces, including porous, tubular, or multiple shapes. The optimal dimension to maximize osseointegration in terms of appearance or distribution on implant surfaces is still a matter of intense research [17].

Several methods can be applied to modify the implant surface to reproduce ECM characteristics. Those methods may be classified as subtractive and additive [24,55,92]. In subtractive methods, the biomaterial is usually removed from the implant surface by anodizing, sandblasting and/or acid etching. Additive methods are those where other materials are added to the implant surface, e.g., plasma spray, hydroxyapatite or calcium phosphate coatings or ion deposition [92,98,99].

### 5.1. Biomimetic Surface Modifications—Additive Manufacturing

#### 5.1.1. Plasma Spray

Plasma spray is a typical additive modification used on titanium surfaces (titanium plasma spray—TPS) that increases the surface roughness through hydroxyapatite deposition [30,98]. In this technique, the particles are injected into a plasma torch at high temperatures, projected onto the implant surface, and allowed to condense and merge. To ensure excellent durability of the coating, the surface is usually sandblasted, and the final coating obtained can range in thickness from a few micrometers to millimeters. It can also be used to obtain surface roughness with Sa > 2 μm [19,30]. Some clinical complications associated with surfaces, such as delamination and marginal bone resorption, have been reported [98,100]. Today there is a consensus on the clinical benefits of using moderately rough implants instead of plasma-sprayed surfaces [19].

#### 5.1.2. Addition of Bioactive Components

The chemical properties of biomaterial surfaces play an essential role in cell-biomaterial interaction and consequently in the osseointegration process. There is a growing concern about bacterial colonization and biofilm formation on dental implants, leading to the development of new implant materials and antibacterial implant surfaces [69,101]. The addition of bioactive components to implant surfaces can be classified into two groups: one that favors cell adhesion and the osseointegration process and the other that decreases bacterial adhesion and biofilm formation [24,83,102,103].

The addition of fluoride, silver, zinc, copper and nickel particles has been suggested by several authors based on their antibacterial properties. Fluoride nanoparticles appear to have the ability to reduce bacterial colonization on the YTZP implant surface [104]. At the same time, silver, zinc, copper, and nickel have been incorporated into titanium surfaces at the level of nanotubules generated by anodization to obtain a surface with antimicrobial activity [105]. Several synthetic and natural bioactive agents have been added to the biomaterial surfaces to enhance bone healing, osseointegration, and implant integration into the peri-implant tissue [19,106]. Hydroxyapatite (HA) or beta-tricalcium phosphate (βTCP) are used as a biological layer of apatite coating that has shown good results in terms of biocompatibility, osteoblast differentiation and osseointegration. However, an in vitro study suggests that bioactive-modified titanium and zirconia surfaces reduce fibroblast cell adhesion, viability, and proliferation compared to pure biomaterial [96,107]. These layers showed low tensile strength (<51 MPa) and fracture toughness (0.28 to 1.41 MPa.m^1/2^). Scientists have developed a new coating method inspired by the natural biomineralization process to avoid these drawbacks. In this process, calcium phosphate crystals deposited on the titanium surface from simulated body fluids (SBF) form a coating at room temperature [108,109].

Several authors have studied the incorporation of growth factors such as transforming growth factor-beta (TGF-ß) and bone morphogenetic proteins (BMPs) as osteoinductive materials over the years. However, although some promising results have been obtained in the bone healing process, the main limitation is their instantaneous and non-progressive release. [110,111].

### 5.2. Biomimetic Surface Modifications—Subtractive Manufacturing

#### 5.2.1. Anodizing

The titanium surface can be modified by an anodization technique using strong acids such as sulfuric acid (H_2_SO_4_), phosphoric acid (H_3_PO4), hydrofluoric acid (HF) or nitric acid (HNO_3_), which increases surface roughness and oxide layer formation [112,113]. Nowadays, one of the most popular brands uses this type of surface treatment (TiUnite, Nobel Biocare, Sweden). In animal and human studies, a higher BIC was observed for dental implants with this type of surface treatment compared to machined implants [112,114].

#### 5.2.2. Blasting and/or Acid Etching

These subtractive procedures can be performed separately or simultaneously. Sandblasting with titanium oxide or alumina particles is another method that can increase surface roughness. Normally, the particles are thrown through an high-speed outlet nozzle [55,92]. Strong acids such as HF, HNO_3_, H_2_SO_4_ or HCL are used to remove oxide impurities. After acid etching, the surfaces are minimally rough with Sa values < 1 μm and a modification of the chemical composition of surfaces [19,92].

Combined sandblasting and acid etching (SBAE) is often used to modify implant surfaces. It involves sandblasting with alumina or titanium particles, followed by acid etching. The main reason for combining these methods is to create a surface with excellent roughness for mechanical fixation and with increased potential for protein adhesion [115]. When comparing implants processed with different surface treatments, SBAE implants showed a greater resistance against reverse torque. [116]. It is noteworthy that this combination, commercially supplied as SLA (Large-grit Sandblasted Acid-Etched) (Straumann, Basel, Switzerland), has shown increased osteoblastic differentiation in vitro compared to smooth surfaces [117,118,119]. However, most of the reported literature is based on surface modifications carried out on titanium, which are currently still poorly described on zirconia.

### 5.3. Biomimetic Surface Patterning

Implant surface patterning by performing biologically-inspired topographies has been described for many years. It was originally used on titanium surfaces to control epithelial cell migratory behavior [120,121]. Over the years, it has been described in the literature as a promising approach to generating directed physical signals, cell regulation, and collagen matrix alignment [122]. This demonstrates a clear relationship between topographical surface patterning, cell behavior, and adhesion [56], with surface topography considered a critical determinant of human cell adhesion, proliferation and differentiation [57,123]. Furthermore, it is reported that smooth surfaces favour fibroblast adhesion, and soft tissue growth, while rough surfaces can be considered enhancers of osteoblast adhesion and bone proliferation. Grooved and microtextured surfaces can provide orientation and directional cues through a phenomenon known as “contact orientation” for osteoblast morphogenesis in the preferred direction [65,124]. One of the first studies to compare and analyse cell behavior in a predefined topography is an in vitro and in vivo study by Chehroudi and co-workers, in which they concluded that at grooves depth of 10 μm, there is a greater number of epithelial cell adhesions than in smooth titanium surfaces [121]. Numerous studies have since been published, but most of them relate to titanium surfaces, with few references related to zirconia surfaces. The studies are contradictory as to the best shapes and dimensions of these macro patterns. [125,126,127]. Several techniques have been used to produce standardized implant surfaces [55], e.g., milling and laser technology [128].

#### 5.3.1. Milling

Milling is a cutting technology aimed at modifying the original shape of the material and knowing its machinability. Milled surfaces can be influenced by cutting tools, the level of surface finish, the type of chips formed, the machining power required for the process, and the volume of chips removed per time interval, among others. These imperfections along milled surfaces allow osteogenic cells to attach and deposit bone, creating a bone-to-implant interface. There are several types of milling cutting tools, including the drill or milling cutter. Milling is an efficient and versatile machining process widely used on an industrial scale. Currently, holes or pores can also be processed with this technique. There is, however, conflicting evidence on the effects of these treatments on cell behavior. Milling topography does not appear to affect hard and soft tissue cell proliferation, but it does appear to influence osteoblast cellular differentiation [129].

#### 5.3.2. Laser

There are several types of lasers, including solid-state lasers (Nd: YAG, Ti: sapphire and fiber laser), gas lasers (CO_2_ and excimer), liquid lasers (organic dye liquid), semiconductor lasers (quantum cascade laser and diode laser) and free-electron lasers (FEL). Lasers can emit in continuous mode, where continuous light is emitted, or in pulsed mode, where there is a period in which light slides between two successive pulses [130].

Nd: YAG and CO_2_ lasers can be used to imprint textures on surfaces and are normally used for zirconia surface standardization. Laser technology uses high-density energy by aiming the laser source at materials to heat, melt, sublimate and modify them at high temperatures, promoting surface texture at macro, micro and nano-level [17,128,131,132]. It increases the zirconia implant’s surface roughness without affecting the crystalline tetragonal phase and keeping the surface clean and homogeneous [50,133]. Laser surface treatment is the only method that allows the rapid and accurate creation of grooves without direct contact with the implant and without risk of contamination [11,128].

This treatment has been described as showing potential in increasing osseointegration. In human studies, titanium dental implants with laser grooves show a reduction in crestal bone loss and inhibition of apical migration of epithelium with a strong attachment of gingival tissue [56]. However, there is less scientific support regarding cell colonization in treated zirconia surfaces textured by different techniques and with different patterns. In vitro studies demonstrate that Nd: YAG laser-patterned zirconia surfaces have superior cell behavior compared to untextured surfaces and also demonstrated that the ideal pattern dimension would be between 10 μm to 1 mm [125].

## 6. Conclusions

This review summarizes the most relevant literature on the role of implant surfaces in biological integration success. Several surface properties affect biological responses, determining the interaction between implant, proteins and cells, namely surface chemistry, roughness, and topography. A variety of techniques and materials that have been used to improve implant surface properties were explained and discussed, including physical and chemical functionalization. Subtractive methods of functionalization involving laser treatment and the embedding of bioactive or antibacterial nanoparticles have produced promising results. Additionally, despite the challenges regarding the immobilization and controlled release of growth factors or other biomolecules over time, their inclusion in the implant surface is another potential approach to biological functionalization. However, there is great heterogeneity between studies considering surface characterization methods, cell culture conditions and cell types, which makes the comparison between the studies challenging. To date, there is no clinical evidence supporting the superiority of one implant surface over the others. Future research should aim at standardizing characterization and defining the critical variables for clinical success. This will enable the development of biologically-inspired surfaces to promote successful and functional integration with surrounding tissues.

## Figures and Tables

**Figure 1 biomimetics-07-00074-f001:**
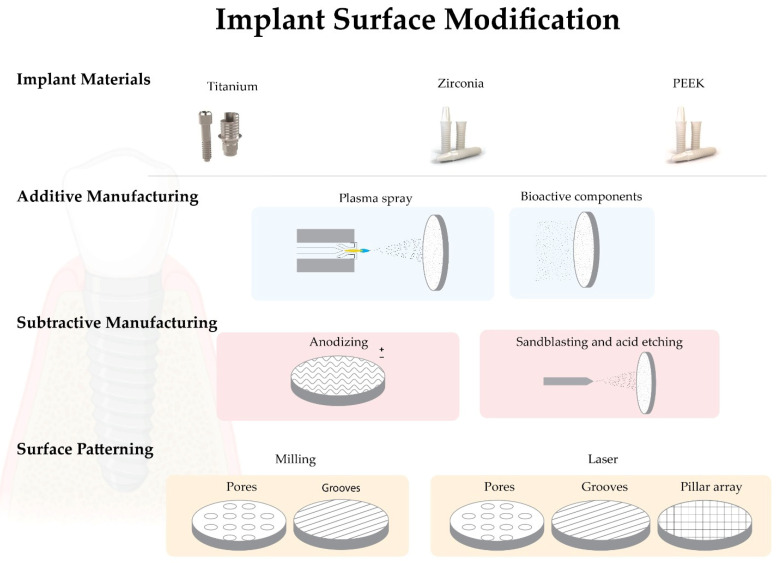
Schematic image of the most relevant biomimetic surface modification strategies.

## Data Availability

Not applicable.

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
