# Peer review of "Biomimetic Implant Surfaces and Their Role in Biological Integration—A Concise Review"

_biomimetics, 2022, doi:10.3390/biomimetics7020074_

Round 1

Reviewer 1 Report

The Authors wrote a concise but content-rich review. This is not a systematic review, but since the topic is relevant and interesting, even a narrative/concise review can be published in Biomimetics.

1. Minor spell check is required throughout the text; moreover, the references in the text are not in correct format/position.

2. The abstract should be structured.

3. The Authors should insert the Aims of the paper in the Abstract and in the Introduction.

4. The review should contain a Methods section which describes the search strategy and inclusion/exclusion criteria.

5. The Authors should summarize the literature search strategy in the Abstract.

6. The references are generally updated, but their number is high. The Authors should consider reducing the number of papers in the References section, including those in which they are co-authors and according to the inclusion/exclusion criteria of the search strategy described in the Methods section.

Reviewer 2 Report

Dear Authors,

I have read the manuscript with interest and some questions raised. Enlisted please find my comments.

Overall. General English grammar revision (Minor spelling errors).

Key words. “dentistry” and “oral surgery” could be added in my opinion.

Abstract. Provide a structured summary including, as applicable: background; objectives; data sources; study eligibility criteria, participants, and interventions; study appraisal and synthesis methods; results; limitations; conclusions and implications of key findings.

Overall. Please follow PRISMA Guidelines for reviews.

Introduction. Authors stated “The first documented dental implant dates back to 600 AD. Since then, various materials have been tested as dental implants, including gold, silver, porcelain, and other materials. ”. Please add a reference for this statement.

Methods. Please add a table showing the risk of bias with a quality assessment of the main studies taken into account to perform revision. Please use the three codified colors to identify low (green), moderate (yellow) or high (red) risk of bias. Please add details about how this information is to be used in any data synthesis.

Please give numbers of studies screened, assessed for eligibility, and included in the review, with reasons for exclusions at each stage, ideally with a flow diagram.

Section 2. Authors stated “Bone is a complex connective tissue composed of a mineralized matrix (30%), which provides mechanical strength, and an organic matrix (70%), providing elasticity and flexibility ”. Please add a reference for this statement.

Section 3. Authors stated “Scientific terms such as osteoinduction and osteoconduction involved in processes related to osseointegration are commonly found in the literature ”. Please add a reference for this statement.

Section 3.2. Please check title spelling.

Section 4. Authors stated “Therefore, the hydrophilic properties promote protein and cell adhesion, interactions of body fluids and may increase tissue healing kinetics. (81–83) ”. Some discussion about microbiology aspects and healing should be added. It could be stated that “Some unexplored variables can have a significant influence on oral environment. The use of probiotics (Butera A, Gallo S, Maiorani C, Preda C, Chiesa A, Esposito F, Pascadopoli M, et al. Management of Gingival Bleeding in Periodontal Patients with Domiciliary Use of Toothpastes Containing Hyaluronic Acid, Lactoferrin, or Paraprobiotics: A Randomized Controlled Clinical Trial. Applied Sciences. 2021; 11(18):8586. ), postbiotics (Butera A, Gallo S, Pascadopoli M, Taccardi D, et al. Home Oral Care of Periodontal Patients Using Antimicrobial Gel with Postbiotics, Lactoferrin, and Aloe Barbadensis Leaf Juice Powder vs. Conventional Chlorhexidine Gel: A Split-Mouth Randomized Clinical Trial. Antibiotics. 2022; 11(1):118) and natural compounds (Antibacterial Properties of Aloe vera on Intracanal Medicaments against Enterococcus faecalis Biofilm at Different Stages of Development. Ghasemi N, Behnezhad M, Asgharzadeh M, Zeinalzadeh E, et al. Int J Dent. 2020 Dec 28;2020:8855277.)can modify oral Clinical and Microbiological Parameters and they could have an effect also in bacterial adhesion to implant surfaces. All these variables should be considered in future clinical trials.”.

Discuss limitations at study and outcome level (e.g., risk of bias), and at review-level (e.g., incomplete retrieval of identified research, reporting bias).

Please provide a general interpretation of the results in the context of other evidence, and implications for future research.

Describe sources of funding for the review and other support (e.g., supply of data); role of funders for the review.

Figures. None presented. Please add flow diagram of studies screened

Tables. None presented. Please add one or more tables showing risk of bias assessment of studies screened.

Reviewer 3 Report

Title: Biomimetic implant surfaces and their impact in biological responses – a concise review

The topic of this article may be clinically and scientifically interesting for different specialists but some suggestions may be considered as follows:

Title

  • The title should be clearer. I suggest to modify the title in “Biomimetic implant surfaces and their role in biological integration – a concise review”

Abstract

  • The abstract is too short. It should also be more informative and specific.
  • Please, add more information about the search strategy.
  • Please, clearly define the aims of this review.

Keywords

  • The keywords “roughness”, “chemical composition” and “cells behavior” should be removed.
    I suggest to add these keywords “dental implants”, “oral rehabilitation” and “Biomimetic”.

Introduction

  • The introduction should be thorough. For example, add more information about why an implant is required.
  • Osseointegration should be better explained or marge the introduction section with the section “Peri-implant bone healing and osseointegration”.
  • Please, move the reference no.8 at the end of the sentence: “Professor Per-Ingvar Brånemark study in 1950 demonstrated that titanium structures could be permanently incorporated into bone such that this interface could only be separated by fracture”.
  • I suggest to insert these references:

    1.Bennardo F, Barone S, Vocaturo C, Nucci L, Antonelli A, Giudice A. Usefulness of Magnetic Mallet in Oral Surgery and Implantology: A Systematic Review. J Pers Med. 2022 Jan 14;12(1):108. doi: 10.3390/jpm12010108. PMID: 35055423; PMCID: PMC8781210.
    2. Contaldo M, De Rosa A, Nucci L, Ballini A, Malacrinò D, La Noce M, Inchingolo F, Xhajanka E, Ferati K, Bexheti-Ferati A, Feola A, Di Domenico M. Titanium Functionalized with Polylysine Homopolymers: In Vitro Enhancement of Cells Growth. Materials (Basel).
    2021 Jul 3;14(13):3735. doi: 10.3390/ma14133735. PMID: 34279306; PMCID: PMC8269806.

Paragraph “2. Peri-implant bone healing and osseointegration”

  • The references no. 16-17 should be moved at the end of the sentence “due to the interactions between the implant, biological fluids and peri-implant tissues”. 

Materials and methods

  • The paragraph “materials and methods” should be added in order to explain the search strategy and which database were used. Also, you should clarify how many articles have been found and how many of them have been selected.

Conclusion

  • Further statements should be added in the conclusions to better understand the results of this concise systematic.

References

  • The articles should be numbered and cited in the text with related numbers.

Language

  • The English language should be improved; a native speaker should revise the manuscript before resubmission.

Round 2

Reviewer 2 Report

Dear Authors,

Manuscript has been revised but surprisingly in the version of the manuscript that I have received, many comments have been ignored. Enlisted please find my comments. Please take into careful account all comments.

Overall. General English grammar revision (Minor spelling errors).

Key words. “dentistry” and “oral surgery” could be added in my opinion.

Abstract. Provide a structured summary including, as applicable: background; objectives; data sources; study eligibility criteria, participants, and interventions; study appraisal and synthesis methods; results; limitations; conclusions and implications of key findings.

Overall. Please follow PRISMA Guidelines for reviews.

Introduction. Authors stated “The first documented dental implant dates back to 600 AD. Since then, various materials have been tested as dental implants, including gold, silver, porcelain, and other materials. ”. Please add a reference for this statement.

Methods. Please add a table showing the risk of bias with a quality assessment of the main studies taken into account to perform revision. Please use the three codified colors to identify low (green), moderate (yellow) or high (red) risk of bias. Please add details about how this information is to be used in any data synthesis.

Please give numbers of studies screened, assessed for eligibility, and included in the review, with reasons for exclusions at each stage, ideally with a flow diagram.

Section 2. Authors stated “Bone is a complex connective tissue composed of a mineralized matrix (30%), which provides mechanical strength, and an organic matrix (70%), providing elasticity and flexibility ”. Please add a reference for this statement.

Section 3. Authors stated “Scientific terms such as osteoinduction and osteoconduction involved in processes related to osseointegration are commonly found in the literature ”. Please add a reference for this statement.

Section 3.2. Please check title spelling.

Section 4. Authors stated “Therefore, the hydrophilic properties promote protein and cell adhesion, interactions of body fluids and may increase tissue healing kinetics. (81–83) ”. Some discussion about microbiology aspects and healing should be added. It could be stated that “Some unexplored variables can have a significant influence on oral environment. The use of probiotics (Butera A, Gallo S, Maiorani C, Preda C, Chiesa A, Esposito F, Pascadopoli M, et al. Management of Gingival Bleeding in Periodontal Patients with Domiciliary Use of Toothpastes Containing Hyaluronic Acid, Lactoferrin, or Paraprobiotics: A Randomized Controlled Clinical Trial. Applied Sciences. 2021; 11(18):8586. ), postbiotics (Butera A, Gallo S, Pascadopoli M, Taccardi D, et al. Home Oral Care of Periodontal Patients Using Antimicrobial Gel with Postbiotics, Lactoferrin, and Aloe Barbadensis Leaf Juice Powder vs. Conventional Chlorhexidine Gel: A Split-Mouth Randomized Clinical Trial. Antibiotics. 2022; 11(1):118) and natural compounds (Antibacterial Properties of Aloe vera on Intracanal Medicaments against Enterococcus faecalis Biofilm at Different Stages of Development. Ghasemi N, Behnezhad M, Asgharzadeh M, Zeinalzadeh E, et al. Int J Dent. 2020 Dec 28;2020:8855277.)can modify oral Clinical and Microbiological Parameters and they could have an effect also in bacterial adhesion to implant surfaces. All these variables should be considered in future clinical trials.”.

Discuss limitations at study and outcome level (e.g., risk of bias), and at review-level (e.g., incomplete retrieval of identified research, reporting bias).

Please provide a general interpretation of the results in the context of other evidence, and implications for future research.

Describe sources of funding for the review and other support (e.g., supply of data); role of funders for the review.

Figures. None presented. Please add flow diagram of studies screened

Tables. None presented. Please add one or more tables showing risk of bias assessment of studies screened.

Reviewer 3 Report

Dear Authors, 
I saw the corrections. 
The manuscript can be accepted. 

Best regards,

Ludovica Nucci

Author Response

Thank you